# Microstructural and Mechanical Properties of Alkali Activated Materials from Two Types of Blast Furnace Slags

**DOI:** 10.3390/ma12132089

**Published:** 2019-06-28

**Authors:** Jun Xing, Yingliang Zhao, Jingping Qiu, Xiaogang Sun

**Affiliations:** College of Resources and Civil Engineering, Northeastern University, Shenyang 110819, China

**Keywords:** alkali activated, blast furnace slags, microstructure, mechanical properties

## Abstract

This paper investigated the effect of blast furnace slags (BFS) characteristics on the properties achievement after being alkali activated. The physical and chemical characteristics of BFS were determined by X-ray fluorescence (XRF), X-ray Diffraction (XRD) and laser granulometry. Multi-technical characterizations using calorimetry, XRD, Fourier Transform Infrared Spectroscopy (FTIR), Thermogravimetry (TG-DTG), scanning electron microscope (SEM), nitrogen sorption and uniaxial compressive strength (UCS) were applied to give an in-depth understanding of the relationship between the reaction products, microstructure and BFS characteristics. The test results show that the microstructure and mechanical properties of alkali activated blast furnace slags (BFS) highly depend on the characteristics of BFS. Although the higher content of basic oxide could accelerate the hydration process and result in higher mechanical properties, a poor thermal stabilization was observed. On the other hand, with a higher content of Fe, the hydration process in alkali activated BFS2 lasts for a longer time, contributing to a delayed compressive strength achievement.

## 1. Introduction

Blast furnace slag (BFS), which is an industrial by-product generated from the Fe and steel industry [1], has been thoroughly studied in the cement and concrete fields due to its high reactivity in alkali environments. It has been reported that the replacement of part of Portland cement, varying from 30% to up to 85%, could improve the durability and resistance to early-age cracking [2], produce high strength and performance concrete, and bring environmental and economic benefits together, such as resource conservation and energy savings [3,4]. Numerous studies [5,6,7,8] with a focus on the reduction of CO_2_ emissions and the practical use of wastes have evaluated BFS as a construction material.

Alkali activated materials (AAM), which have been widely discussed as a ‘sustainable cement binder’ [9], could be generated from a wide range of aluminosilicate precursors, such as BFS, fly ash or metakaolin. Compared with Portland cement, AAM enjoys a quick compressive strength development [10], lower permeability [11], and good resistance to acid and fire attacks [12]. Moreover, almost no SOx, NOx, or CO_2_ are generated in the process of AAM preparation [13], contributing to a great interest in the study and development of AAM worldwide.

As one of the commonly used raw materials for AAM preparation, BFS has attracted a lot of studies in the world. Similar to Portland cement, which possesses hydraulic properties due to the large amounts of reactive CaO and SiO_2_ [14], most BFS contains CaO (30–50%), SiO_2_ (28–38%), Al_2_O_3_ (8–24%), and MgO (1–18%), although the specific component depends on the composition of the raw materials in the Fe production process. CaO, SiO_2_, Al_2_O_3_ and MgO have been identified as having a major influence on the BFS reactivity, which could be easily activated using an alkali solution [15]. The properties of AAM may vary depending on the properties of the raw slags. Generally, increasing the CaO content could result in an enhanced basicity [(CaO + MgO)/(SiO_2_ + Al_2_O_3_)] and hydration modulus [(CaO + MgO + Al_2_O_3_)/SiO_2_], contributing to a higher compressive strength and the compact microstructure of the resultant AAM. Meanwhile, an excessive CaO content has been reported to be detrimental to the properties of AAM. The content of MgO also shows a similar advantage to the properties of AAM, and Chen Y et al. [16] reported that the sulfate resistance of AAM increased with the MgO content in slag due to the favored formation of layered double hydroxides (LDHs). Bernal et al. [17] found that when the MgO content was higher than 5%, hydrotalcite was identified as the main product, and a reduced susceptibility to carbonation was observed.

Fe containing precursor materials and the role of Fe during the alkali activation are also currently receiving attention. van Deventer et al. deduced that reactive Fe from fly ash could reprecipitation much faster than Si and Al during geopolymerization, according to the report by Daux et al. Kaze et al. studied the potential by using Fe-rich laterite with kaolinite as precursor materials to synthesize inorganic polymers through alkaline activation. They found that the dissolved Fe^3+^ participated in the reaction of geopolymerization and then was inserted into the network matrix of aluminosilicate hydrate. Hu et al. reported the geopolymer material synthesized from alkali-thermal pretreated Fe-rich Bayer red mud (RM) and fly ash and investigated the role of Fe species in geopolymerization by Mössbauer spectra. They found that the coordinated Fe^3+^ replaced Al^3+^ in the aluminosilicate structure of geopolymer. Kaze et al. developed building materials by applying the technology of geopolymerization using naturally abundant Fe-rich laterite as raw material and rice husk ash. The enhanced mechanical properties of laterites-based geopolymer composites resulting from the formation of Fe-silicates made them promising candidates for applications in construction.

However, there is limited research about BFS containing a high level of Fe and the distinct characteristics after alkali activation. This paper presents the preparation of AAS using two slags (one of them contains a high level of Fe) with various properties. The performances are systematically compared and evaluated in terms of hydration kinetics, phase changes, an FTIR analysis, thermal and mechanical properties, as well as the microstructure.

## 2. Experimental Section

### 2.1. Materials

Two types of BFS were obtained from two steel factories in Hebei, China and were used as source materials to synthesize AAS. The chemical composition of the slags determined by XRF is given in Table 1. The SiO_2_ content in BFS1 and BFS2 was 32.53% and 31.12%, indicating no significant difference. Meanwhile, the Al_2_O_3_ content in S1 was 16.12%, approximately one times higher than BFS2. Similarly, BFS1 also contained higher CaO (38.01%) and MgO (8.71%) than BFS2. However, the Fe_2_O_3_ content in BFS1 was only 1.54%, while this number was 34.94% in BFS2. On the other hand, BFS1 enjoyed a higher basicity index (0.96, Table 1) than BFS2 (0.31), indicating that BFS1 had a higher hydraulic potential, because the CaO content in the slag controlled the activation. This could be concluded from the number of the hydration modulus (Table 1).

The particle size distributions of the BFS were shown in Figure 1 and Table 2. The particle size distribution is reported to be one of the factors that strongly affects the reactivity of raw materials. With no significant difference, the mean particle diameters (Table 2) of BFS1 and BFS2 were 17.9 and 19.0 μm, respectively. The BET surface areas of BFS1 and BFS2 were 905.3 and 843.4 m^2^/kg, respectively.

### 2.2. Sample Preparation

Alkali activated blast furnace slags (AAS) samples were prepared by mixing BFS and NaOH solution (6 M) at a NaOH solution/slag mass ratio of 0.5 for 5 min in a mixer. The mixtures were then cast into molds, which were sealed with plastic bags to prevent water evaporation, and cured at 60 °C for 24 h in an oven. After that, all the samples were kept in a standard curing chamber with a relative humidity of 90% at 20 °C until the testing date. The labels ‘AAS1′ and ‘AAS2′ in this paper represented the AAS samples made from two types of slag, ‘BFS1′ and ‘BFS2′, respectively.

### 2.3. Characterization

#### 2.3.1. Calorimetry

A hydration kinetics analysis of AAS was assessed by isothermal calorimetry experiments (MicroCal VP-ITC, Malvern Panalytical, Malvern, UK) under a constant temperature of 20 °C. The paste samples with a NaOH solution/slag mass ratio of 0.5 were tested for 40 h with a resolution of 1 min.

#### 2.3.2. Compressive Strength

To determine the compressive strength of the AAS samples, uniaxial compressive strength (UCS) tests were performed by using a computer-controlled mechanical press with a loading capacity of 50 kN and a displacement loading speed of 1 mm per minute, according to the ASTM D2166/D2166M-16 standard [18]. For each mix formulation, triplicate tests are conducted and the average values are recorded.

The fragment after UCS, which was used for the following tests, was impregnated using acetone for 24 h to stop the hydration process and then dried under vacuum at 40 °C.

#### 2.3.3. X-Ray Diffraction (XRD)

The treated fragments in Section 2.3.2 were grounded to a fine powder. An x-ray diffraction (XRD) analysis was carried out using an X’ Pert Pro XRD (Philips, Holland) at a scanning rate of 0.1 deg s^−1^ in the 2θ range of 10 to 80°.

#### 2.3.4. Fourier Transform Infrared Spectroscopy (FTIR)

Fourier-transform infrared spectroscopy (FTIR) tests were performed by NEXUS 470 in America with a wavenumber of 4000–400 cm^−1^ to identify the functional group of the materials.

#### 2.3.5. Thermogravimetric Analysis (TG)

TD-DTG analyses were performed using Diamond TG-DSC (Netzsch STA 409 PC/PG, Selb, Germany). The temperature was raised up from 25 to 1000 °C at a heating rate of 10 °C/min in an alumina crucible under N_2_ atmosphere.

#### 2.3.6. Scanning Electron Microscopy (SEM)

Microtopography was determined by scanning electron microscopy (SEM) using a Phillip XL 30 SEM (Phillips, Holland) with an accelerating voltage of 20 kV electrical pulses for the characterization of the surface morphology. The samples described in Section 2.3.2 were coated with gold. SEM was performed with an accelerating voltage of 20 kV electrical pulses to ensure a good resolution of the images.

#### 2.3.7. Nitrogen Sorption Test

The fragment in Section 2.3.2 was crushed to 4–5 mm diameter and a Nitrogen sorption test was carried out to analyze the pore structure of the AAS sample, using the Automated Surface Area and Porosimetry System (Micrometrics ASAP 2020, France) at a constant temperature of 20 °C.

## 3. Results

### 3.1. Hydration Kinetics

The heat release curves of AAS1 and AAS2 are shown in Figure 2. As seen in Figure 2, an initial pre-induction period in the first hour is observed, followed by a heat release peak, which is consistent with the previous results of the initial decomposition of slag [19]. This period was then followed by a long induction period with little heat released, which was also reported in the literature [20]. However, compared to AAS1, AAS2 enjoys a longer induction period (about 20 h), observed here before the precipitation of reaction products starts taking place, while the same period for AAS1 is about 5 h. This can be attributed to the lower basicity index and hydration modulus, which are listed in Table 1.

From Table 1, BFS1 contains a higher amount of basic oxides, such as CaO and MgO, which could result in a faster polymerization of hydration gels. The reaction between water and CaO could generate Ca(OH)_2_, an acceleration effect on the alkalinity of the reaction system. In the alkali-activated process, OH^−^ was found to have a significant effect on both the compressive strength and structure of the alkali-activated materials [21]. Leaching of Si^4+^, Al^3+^, Ca^2+^ and other minor ions begins when the raw materials come into contact with OH^−^, and the amount of leaching is dependent on the OH^−^ concentration [22]. On the other hand, the hydration products caused by CaO and MgO could magnify the dissolution of BFS, rendering a driving force for the hydration process [23].

### 3.2. XRD Analysis

An X-ray diffraction analysis was carried out to investigate the different phases present in the BFS and AAS. The XRD data for both the AAS1 and AAS2 samples were collected after curing for 7 days and 28 days. The XRD patterns of the BFS are shown in Figure 1, which reveals their amorphous nature, as reflected by the presence of a wide band located approximately at 2θ = 30°–40°. However, it was obvious that there were differences in the mineral composition of the initial slags. The spinel phase, whose formation was a function of both the chemical composition of the melt and its cooling rate [23], was the only crystalline mineralogical phase that was detected in BFS2. As for BFS1, the quartz, calcite and gehlenite phases were detected.

In the XRD pattern of AAS1 (Figure 3a), the quartz and gehlenite phases remained after the alkali activation as these phases were difficult to dissolve in the alkali solution. The strong peak located at around 2θ = 30° was the C–S–H phase, which was one of the main products of alkali activated slag [24,25]. The semi-amorphous nature of C–S–H resulted in the background hump in the 2θ range of 25–35°. A relatively lower intensity peak at 2θ = 11.5° was also observed, at both 7 days and 28 days samples, and was assigned to the hydrotalcite phase. The hydrotalcite phase was usually detected in the alkali activated slag when there was enough MgO content [26]. Traces of calcium carbonate were also identified in the AAS1 samples, which was likely a result of the absorption of carbon dioxide from air when the samples were analyzed [19].

The only crystalline mineralogical phase of BFS2 (Figure 3b) had not undergone any substantial change during the alkali activation process. However, the diffused peak between 2θ = 30–40° had slightly shifted to higher values after the alkali activation, indicating the partial dissolution of the amorphous phase and the formation of a new amorphous phase [27]. As Prof. Joseph Davidovits reported, this amorphous phase enjoyed a molecular structure where part of the Fe atoms was observed as tetrahedral Fe [IV] in a structural position within the Ferro-sialate geopolymeric sequence [–Fe–O–Si–O–Al–O–]. In a previous study, Maragkos et al. [27] also found a similar phase change during alkali-activated ferronickel slag which contained high Fe. They also reported a new peak located at 2θ = 5°, attributed to an amorphous zeolitic phase, which is not noticed in the present study. This difference from the present study may due to the dissimilar characters of the slag and the reaction conditions. In the XRD pattern of AAS2, a small C–S–H peak was detected. This could be explained by the low CaO content in BFS2, and the fact that only a trace of C–S–H was generated. Also, the C–S–H peak might be overlapped with other peaks.

On the other hand, for AAS1, there were no clear differences between the XRD patterns of the 7 days and 28 days samples, which indicated that the hydration reaction of BFS1 was almost completed in the first 7 days and reacted slowly after that. Furthermore, the quick setting (less than 30 min) of AAS1 was also found in the process of the experiments. As for AAS2, the intensity of the peak between 2θ = 30–40° increased in the 28 days samples compared with the 7 days samples. This phenomenon showed that the hydration reaction of BFS2 in the alkali solution remained sharp until 28 days and might keep up in the following days (this was not shown in the present study). The substantial growth in the compressive strength (see Figure 8 in Section 3.6) of AAS2 from 7 days to 28 days was believed to result from the continuous reaction.

### 3.3. FTIR Analysis

Figure 4 represents the FTIR-spectra of BFS1, BFS2, AAS1 and AAS2 (28 days). The FTIR spectra for BFS1 and BFS2 contain two wide and intense bands. The spectrum for the raw slags showed a broadband characteristic of gehlenite, which lies between 900 and 1100 cm^−1^, attributed to the asymmetric stretching vibration of the Si(Al)–O–Si bonds [28]. Another peak, centered at about 490 cm^−1^, is assigned to the bending vibration of the O–Si–O bonds [29]. A weak peak related to CO_3_^2−^ at around 1435 cm^−1^ is also observed. It is obvious that in the spectrum of BFS1 this peak is more intense than BFS2. After being alkali activated, the main absorption band that is related to a Si–O (Al) anti-symmetric stretching vibration at 960 cm^−1^ becomes more intense, with a decrease in the width of the bands at 7 and 28 days, respectively. This phenomenon may result from the increase of the crystallinity of the AAS sample caused by the ordering of the structure. Similar results have been reported by other studies [28,29,30]. It should be highlighted that more obvious changes could be found in the spectrum of AAS2 (Figure 4b) than AAS1 after the alkali activation (Figure 4a). This fact should be related to the rapid reaction of alkali-activated BFS1 and to the fact that there was no significant increase of the hydrated compounds after 7 days. In the XRD analysis, similar results have been concluded. Absorption bands in the 1650 region are related to the bending (H–O–H) and stretching of (OH)^–^ groups, respectively. The intensity of the bands increased from 7 to 28 days, especially for AAS2, due to the continuous activation and formation of hydrated products, such as N–A–S–H, C–S–H, C–A–H and C–A–S–H [29,30].

### 3.4. Thermal Properties

The thermogravimetric results show different trends in the weight loss of the AAS materials as reported in the TG and DTG curves in Figure 5. Normally, in alkali-activated materials the weight loss is characterized by several weight loss changes: the first stage is the loss of the adsorbed or interlayer water of C–S–H [4,31] and the decomposition of ettringite [32] between 30 and 200 °C, and, between 200 and 600 °C, the dehydroxylation process of the gel phase such as C–S–H and C–A–S–H [33,34]. The weight loss after 600 °C is believed to be the result of the viscous sintering process of the alkali-activated materials matrix [31] and the decomposition of anhydrite [34]. In general, the weight loss percentages of AAS1 and AAS2 are 14.9% and 3.6%, which means that AAS2 enjoys a better thermal stability at a high temperature. From Figure 5b, large peaks appeared from 25 to 200 °C in the DTG curves due to the decomposition of ettringite and the loss of the adsorbed or interlayer water of C–S–H [32]. It is obvious that this peak in AAS1 is more intense than AAS2, indicating that more C–S–H product is generated in AAS1. A similar result has been concluded from the XRD analysis, which shows that there is more C–S–H phase in AAS1. In the XRD analysis, a hydrotalcite (Mg_4_Al_2_ CO_3_(OH)_16_·4(H_2_O)) phase in 2θ = 11.5° is observed, which is due to the higher content of MgO in BFS1. Hydrotalcite has a layered double hydroxide structure like brucite, in which the interlayer region contains CO_3_^2−^ ions and water molecules [35,36]. Hydrotalcite decomposes when heated to a certain temperature. When the air temperature is below 200 °C, only interlayer water is lost. When heated to 500 °C, CO_3_^2−^ ions in hydrotalcite completely decompose into CO_2_ [37,38]. This is in agreement with the data in DTG curves in Figure 5b, which show an endothermic peak from 250–500 °C [39]. No similar peak is observed in AAS2, resulting from the lower MgO content. Another broad endothermic peak is detected in the DTG curve of AAS1 from 500–700 °C, which is due to the sintering of the decomposed metal oxides [40]. Hydrotalcite is usually detected in the alkali-activated slag when there is enough MgO content [26], and the appearance of hydrotalcite results in the lower thermal stability of AAS1 compared to AAS2.

A slight weight gain is observed above 700 °C in the TG curve (Figure 5a) of AAS2, associated with an intense endothermic peak in the DTG curve (Figure 5b). A similar result was also reported by Lassinantti Gualtieri [41]. The authors prepared inorganic polymers from laterite using activation with phosphoric acid and an alkaline sodium silicate solution. As no weight increase is found in the TG curve of AAS1 (Figure 5a), this phenomenon possibly results from the higher content of Fe in BFS2, and some reaction took place during the alkali activation process, such as the interaction of Fe with the alkali-activated network [41]. The auto-reduction of Fe during heating may take place, releasing oxygen and turning into ferrous Fe [42]: 4Fe^3+^ + 2O^2−^→4Fe^2+^ + O_2_. Lemougna et al. [43] used ^57^Fe Mössbauer spectroscopy to monitor the behavior of Fe during the synthesis reaction of alkali-activated volcanic ash. The results showed that the augite in the original volcanic ash did not react with the alkali solution but reacted to form new ferric sites, suggesting the possible incorporation of ferric Fe in the tetrahedral network of the geopolymer product. Considering the similarity of the raw materials, ferro-geopolymer networks were possibly also formed in the samples studied here.

### 3.5. SEM Analysis and Pore Size Distribution

Figure 6 illustrates the fracture surfaces of two types of AAS. Cracks occur in the two specimens, but the minimum number of cracks is related to AAS1 (Figure 6a). In general, the microstructure is a heterogeneous matrix consisting of alkali-activated gel and unreacted or partially reacted slag particles, and more unreacted particles are observed in AAS2. As discussed above, BFS1 has a higher content of CaO and Al_2_O_3_, which are the indispensable components during the alkali-activated process [44,45]. Then, after the alkali activation, AAS1 could produce more hydration gel, such as N–A–S–H, C–S–H and C–A–S–H, contributing to a more continuous matrix [46]. The particles are well bridged by the reaction product, which enhances the mechanical strength, in line with the existing literature [47,48]. The coexistence of N–A–S–H gel and C–(A)–S–H gel has reported to be helpful for bridging the gaps between the different hydrated phases and unreacted particles, then contributing to a matrix that is denser and more homogeneous [49].

An EDX analysis was done at multiple points (excluding unreacted particles), and the result is shown in Table 3, based on an average of at least five points. The gel of AAS1 mainly contains Ca, Si and Al with a Ca/Si molar ratio of 1.56, corresponding to the C–S–H gel. This ratio was reported to be about 1.1 in alkali-activated slag [50] and 1.2–2.3 in Portland cement [51]. As has been reported [52], C–S–H gel could coexist with C, M–(A)–S–H and C–(N)–A–S–H gel, resulting from the Na partial substitution by Mg and Al in a Q_4_ framework silicate structure or chain silicate C-S-H type gels [53]. BFS1 contains a higher amount of Ca, along with Mg and Al, which could promote the formation of additional hydrated products, enhancing the physical and mechanical properties of the resultant AAS. For AAS2, a lower Ca/Si ratio (0.47) is observed due to the lower content of Ca in the raw material. Meanwhile, the presence of Fe (28.67%) in the gel and lower content of Al (3.63%) could mean that Al atoms are partially substituted by Fe atoms. The possibility of Fe incorporation in the tetrahedral structure during alkali activation has already been reported by Djobo et al. [47] and Lemougna et al. [43].

It has been reported [54] that there are three classes of pores in the alkali-activated materials matrix: gel pores (2 nm ~ 50 nm), capillary pores (10 nm ~ 1000 nm) and air voids (>1 μm). The pore size distribution of AAS is shown in Figure 7. Generally, the pores in both AAS1 and AAS2 distribute similarly in the range of 10 to 100 nm in diameter, corresponding to the gel pores and the capillary pores. Additionally, it is obvious that AAS1, both for 7 days and 28 days curing, enjoys a smaller pore size and porosity, resulting in a dense and homogenous matrix (Figure 6a,b).

After the alkali activation and 7 days curing, the average pore size of AAS1 is 17.996 nm (Figure 7), and this number turns to 16.641 nm after 28 days curing without a remarkable change. From the analysis of hydration kinetics in Section 3.1, the hydration speed of AAS1 is fast and slow down after 20 h. The XRD analysis (Section 3.2) also shows that there were no clear mineralogy changes between the XRD patterns of the 7 days and 28 days sample, indicating that the hydration reaction of AAS1 was almost completed in the first few days and reacted slowly after that. Similar results could also be concluded from the FTIR analysis (Section 3.3).

When comparing with AAS1, extending the curing time from 7 days to 28 days significantly decreases the average pore size and porosity of AAS2. This could be attributed to the gel filling effect, as the hydration proceeds during the curing period, which is in line with the analyses of the above sections.

### 3.6. Compressive Strength

The compressive strength results of AAS are presented in Figure 8. In general, the compressive strength, both for AAS1 and AAS2, increased with the extension of the curing time from 3 days to 7 days, and AAS1 enjoyed a higher compressive strength throughout the testing time. A relatively low compressive strength (7.58 MPa for AAS1 and 3.86 MPa for AAS2) was observed after 3 days, which results from the incomplete hydration process and fewer gels, like N–A–S–H, C–S–H and C–A–S–H, generated on the one hand, and to the weak bond strength between the unreacted slag particles and binder gel, on the other hand [47]. It would take a long time for the alkali-activated materials to get a higher compressive strength curing at room temperature [55]. After 7 days of curing, the compressive strength increased to 39.45 MPa and 18.59 MPa for AAS1 and AAS2, respectively. This significant increase in the compressive strength could be attributed to the formation of N–A–S–H and C–(A)–S–H gels acting as microaggregates to fill the void in the AAS matrix. The tight interface among these gels was also reported to contribute to the compressive strength [47]. Furthermore, it is obvious that the rate of increase for AAS1 is larger than that of AAS2 from 3 days to 7 days. As has been reported above, BFS1 contains higher reactive oxide components, such as CaO, Al_2_O_3_, and SiO_2_, which could be conducive to the hydration process. This, then, could lead to the generation of more gel phases, resulting in building more compact microstructure and achieving a higher compressive strength. It has been shown that the presence of a sufficient amount of MgO in slag could contribute to the compressive strength improvement of the AAS mortar at early ages, resulting from the formation of more hydration products, such as M–(A)–S–H, a hydrotalcite type gel product, which could refine the porous AAS matrix [56,57]. No remarkable strength increase was observed for AAS1 after 28 days curing, which means the hydration process was almost complete before 28 days, in accordance with the microstructure analyses (XRD, FTIR, TG-DTG and pore size distribution).

It seems that AAS2 needs a longer curing time to acquire a higher compressive strength. After 28 days curing, the compressive strength of AAS2 increased to 33.19 MPa, nearly 50% of the total increase rate during the 28 days, probably due to the higher Fe content. Dileep et al. [58] have attempted to produce Fe geopolymers containing 100% Fe and reported that the resultant gel hardened after 360 days curing. The behavior of Fe may either act as a filler, integrate with the hydration gel or be incorporated into the tetrahedral network to form another gel type. Perera et al. [59] reported that Fe in the octahedral sites in the geopolymer network exists as isolated ions or as oxyhydroxide aggregates. In the hydration process, Fe accumulation led to the formation of nodule nuclides that grew with the Fe dissolution and, finally, gels containing Fe formed with amorphous silica acting as a binder [60].

## 4. Discussion

The alkali activation process starts from the BFS dissolving to release Al^3+^, Si^4+^ and Ca^2+^ after coming into contact with alkali activator solutions, indicated by the exothermic peak in Figure 2. The rate of the formation of hydration products is possibly dependent on the leaching speed. As shown in Figure 2, it takes less time for AAS1 to reach the peak value; on the other hand, AAS1 enjoys a higher exothermic peak compared to AAS2, indicating that the hydration speed of AAS1 is faster. On the one hand, BFS1 contains a higher amount of basic oxides, such as CaO and MgO, which could react with water and generate Ca (OH)_2_ and Mg (OH)_2_, enhancing the amount of OH^−^. In the alkali activation process, OH^−^ could accelerate the leaching speed of Al^3+^, Si^4+^ and Ca^2+^, which are block components of hydration products (i.e., C–(A)–S–H, N–A–S–H). On the other hand, as reported by Van Deventer et al., the active Fe from the BFS2 could reprecipitate as hydroxide or oxyhydroxide phases more rapidly than Al, Si or Ca ions, which will consume the OH^−^ions from the solution phase and therefore slow down the dissolution of the remaining BFS particles.

The compressive strength of AAS highly depends on the amount of hydration product after the BFS has been activated by the alkali solution. As described above, BFS1 contains a higher content of basic oxides, which can accelerate its dissolution and generate more hydration product, mainly as C–(A)–S–H, at an early stage. This could lead to a more heterogeneous matrix in the microstructure (see Figure 7 in Section 3.5). However, the Fe content in BFS2 is higher, hindering the leaching of Al^3+^, Si^4+^ and Ca^2+^ ions, which are the main constituent parts of the hydration product. Therefore, the hydration of BFS2 lasts for a long time, even after 27 days. This is why the growth rate of the compressive strength for AAS2 is larger than that of AAS1 at the later stage (see Figure 8 in Section 3.6). A thermogravimetric analysis has been used to signify the amount of hydration production in many studies. From Figure 5, the mass loss for AAS1 is about 5.4% before 200 °C, mainly resulting from the loss of the adsorbed or interlayer water of C–(A)–S–H and the decomposition of ettringite. This number is larger than AAS2, whose mass loss is approximately 1.9%. The larger weight loss during this temperature period agrees well with the compressive strength testing result (see Figure 8) and microstructure analysis (see Figure 7). The larger formation of C–(A)–S–H and ettringite should be related to higher strengths and a more compact matrix.

Although the compressive strength of AAS2 is lower than AAS1 at any curing age, AAS2 enjoys a better thermal stability. The weight loss percentages of AAS1 and AAS2 are 14.9% and 3.6% after heating to 1000 °C. As has been reported in the XRD analysis (see Figure 3 in Section 3.2), AAS1 contains more C–(A)–S–H, hydrotalcite and calcium carbonate, which do not have a thermal stability at a higher temperature. Dehydration, dehydroxylation and decomposition will take place when heated, leading to a larger mass loss. Meanwhile, by contrast, alkali-activating BFS2 produces fewer such hydration products. On the other hand, BFS2 contains more Fe^2^O^3^, and Fe will inset into the hydration network ([–Fe–O–Si–O–Al–O–]) after being alkali-activated. The EDX analysis (see Table 2 in Section 3.5) also confirms this idea, which shows that the Fe content in the hydration product of AAS2 is 28.67%. Therefore, the results in this study reveal the prospect that BFS contains a higher content of Fe to make alkali-activated materials heat resistant.

In general, although the BFS2 shows a slow hydration rate when being activated by an alkali solution, it is possible to add some accelerator or preparation conditions to enhance the hydration rate. Similar studies are in progress in our research team.

## 5. Conclusions

This study investigated the effect of blast furnace slags (BFS) characteristics on property achievements after being alkali-activated. Two types of BFS with different physical and chemical characteristics, activated with 6 M NaOH solution were used for the experiments, and XRD, FTIR, TG-DTG, SEM, nitrogen sorption and UCS were applied to study the relationship between the reaction products, microstructure and BFS characteristics. The main conclusions from this work are as follows.
(1)The characteristics of the BFS can significantly affect the properties of AAS.(2)Due to the higher content of reactive oxide components, such as CaO, Al_2_O_3_ and MgO, BFS1 enjoys a higher basicity index and hydration modulus, which accelerates the hydration process, accompanying a higher early compressive strength and compact microstructure. However, the thermal stabilization of AAS1 is poor, resulting from the hydration gel composition, which is mainly C–(A)–S–H oriented.(3)The higher thermal stabilization and delayed strength development of AAS2 occurred mainly due to the higher content of Fe in BFS1, which either acts as a filler, integrates with the hydration gel or is incorporated into the tetrahedral network to form another gel type during the alkali-activated process.

## Figures and Tables

**Figure 1 materials-12-02089-f001:**
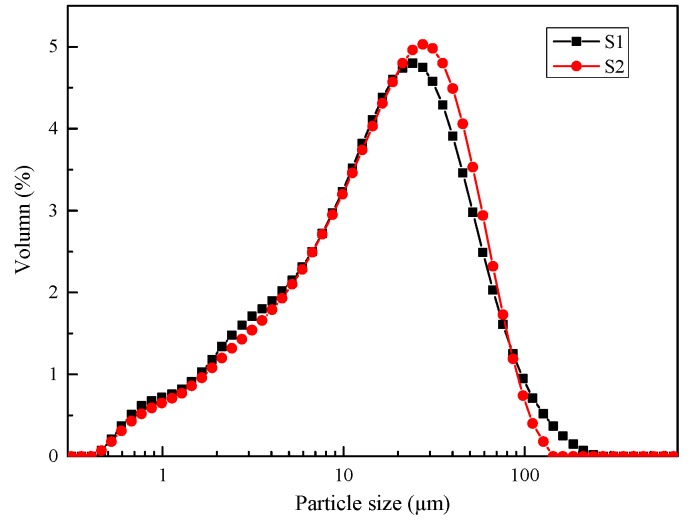
Particle size distributions of the BFS.

**Figure 2 materials-12-02089-f002:**
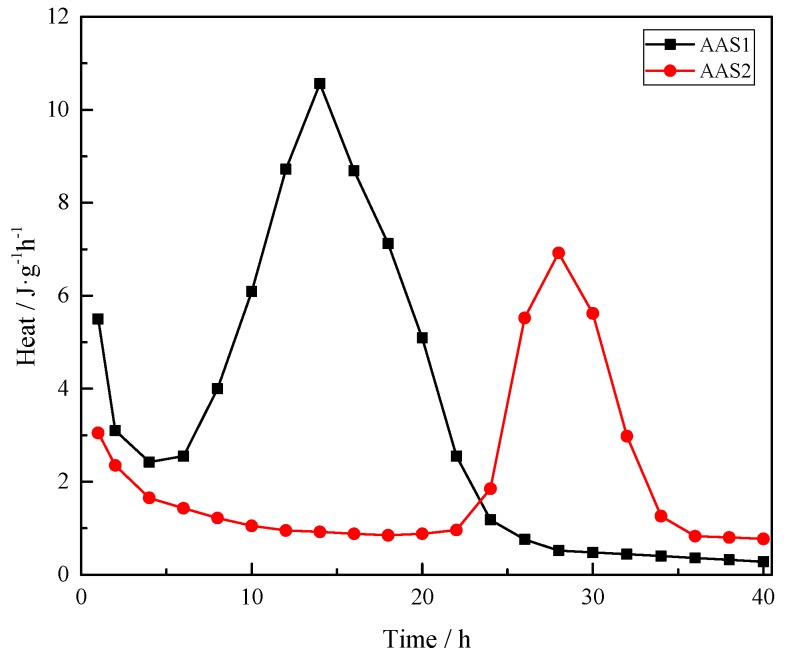
Isothermal calorimetry data for AAS.

**Figure 3 materials-12-02089-f003:**
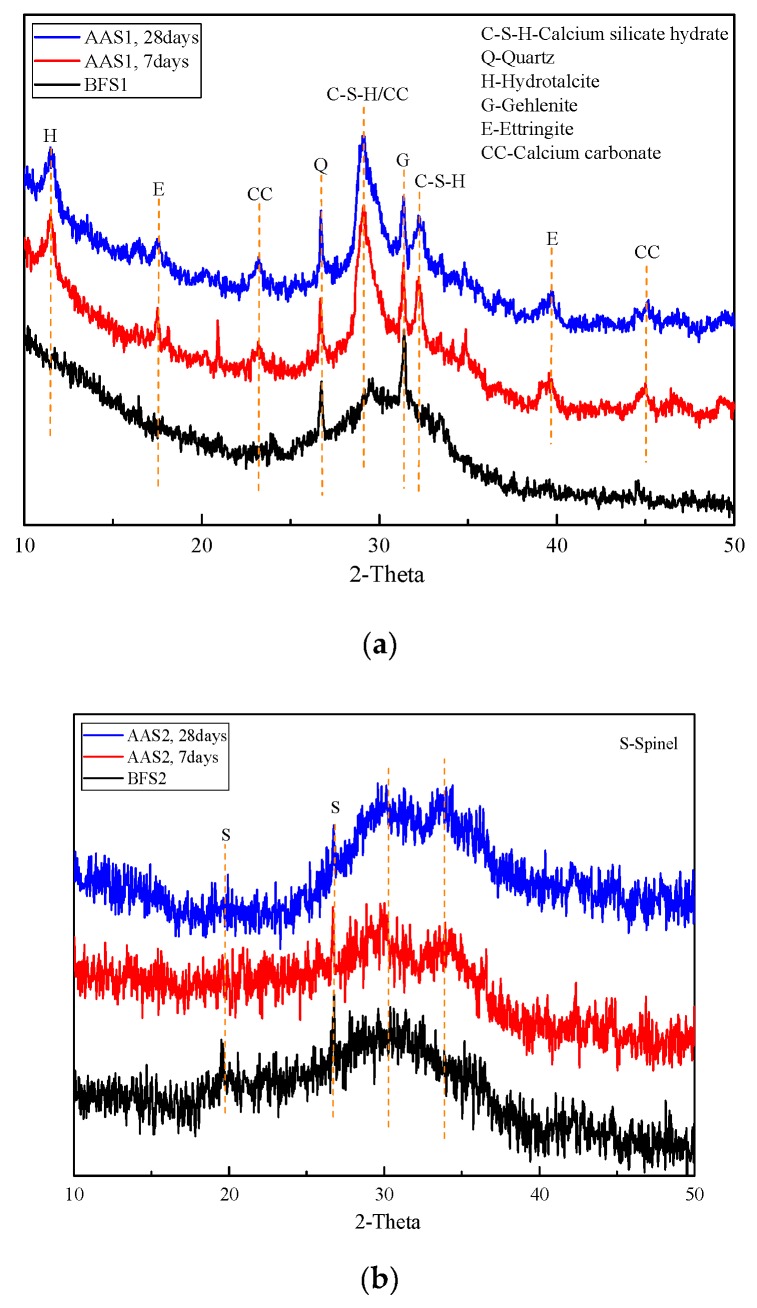
XRD patterns of AAS. (**a**) AAS1 and (**b**) AAS2.

**Figure 4 materials-12-02089-f004:**
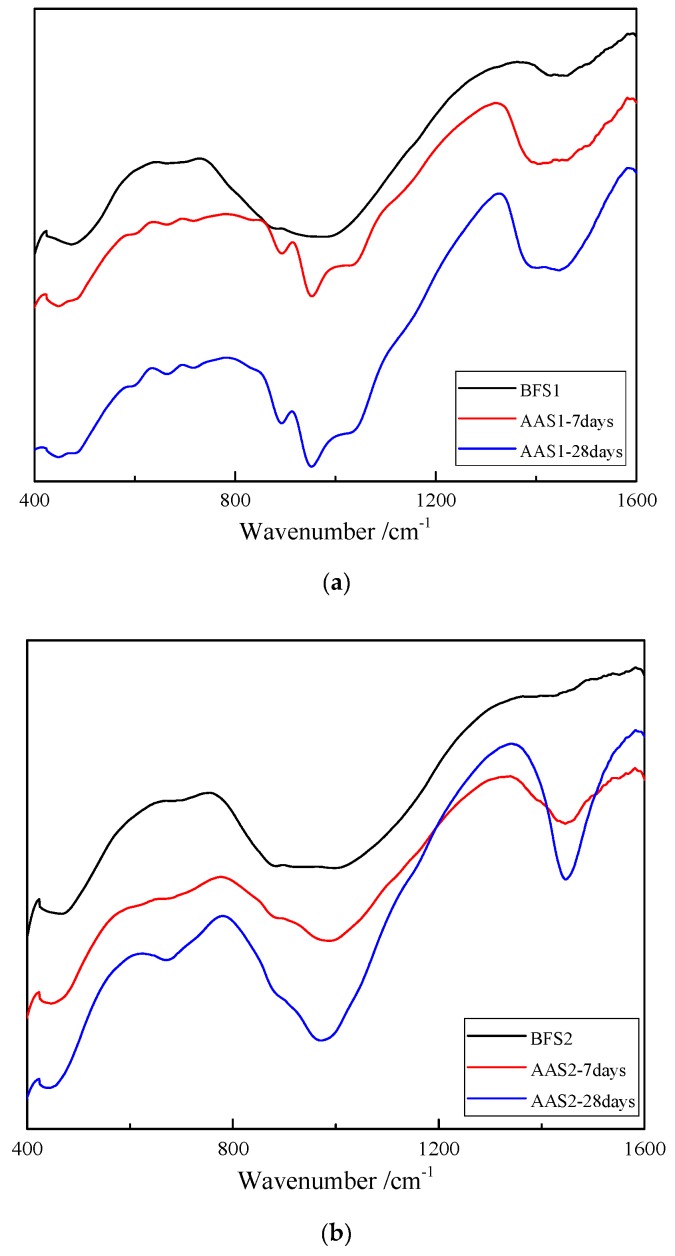
Results of the FTIR analysis. (**a**) AAS1 and (**b**) AAS2

**Figure 5 materials-12-02089-f005:**
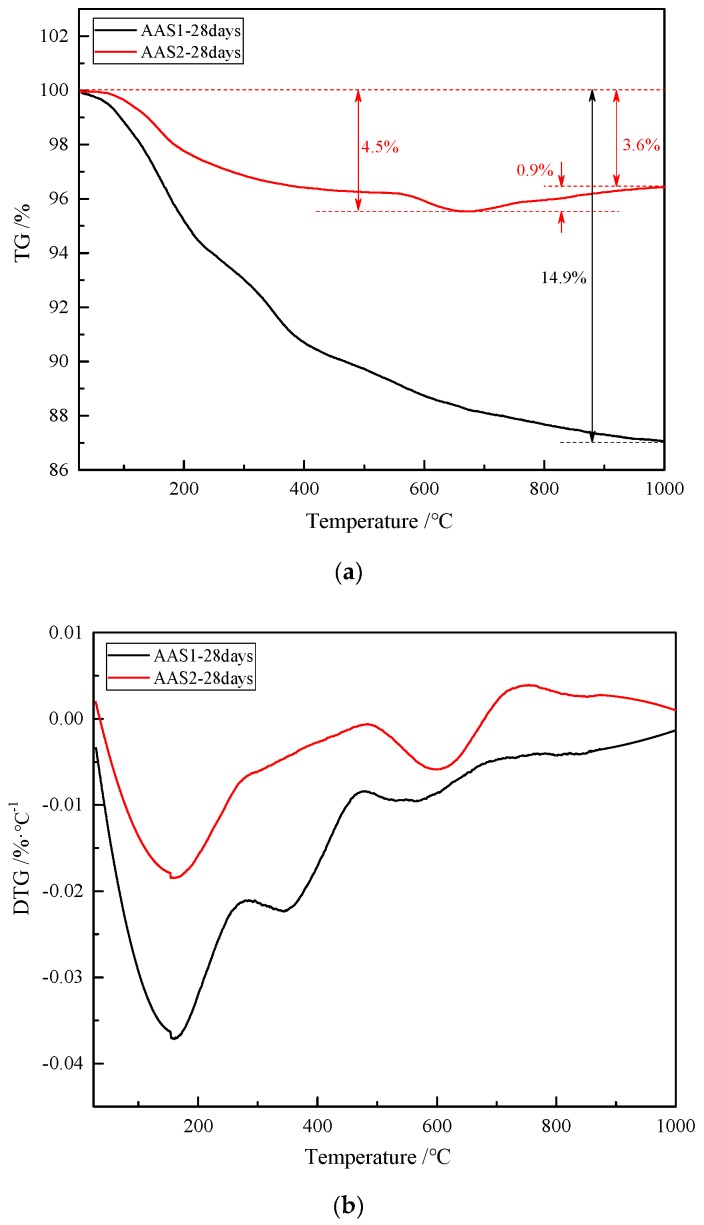
TG and DTG curves of the AAS. (**a**) TG curves and (**b**) DTG curves.

**Figure 6 materials-12-02089-f006:**
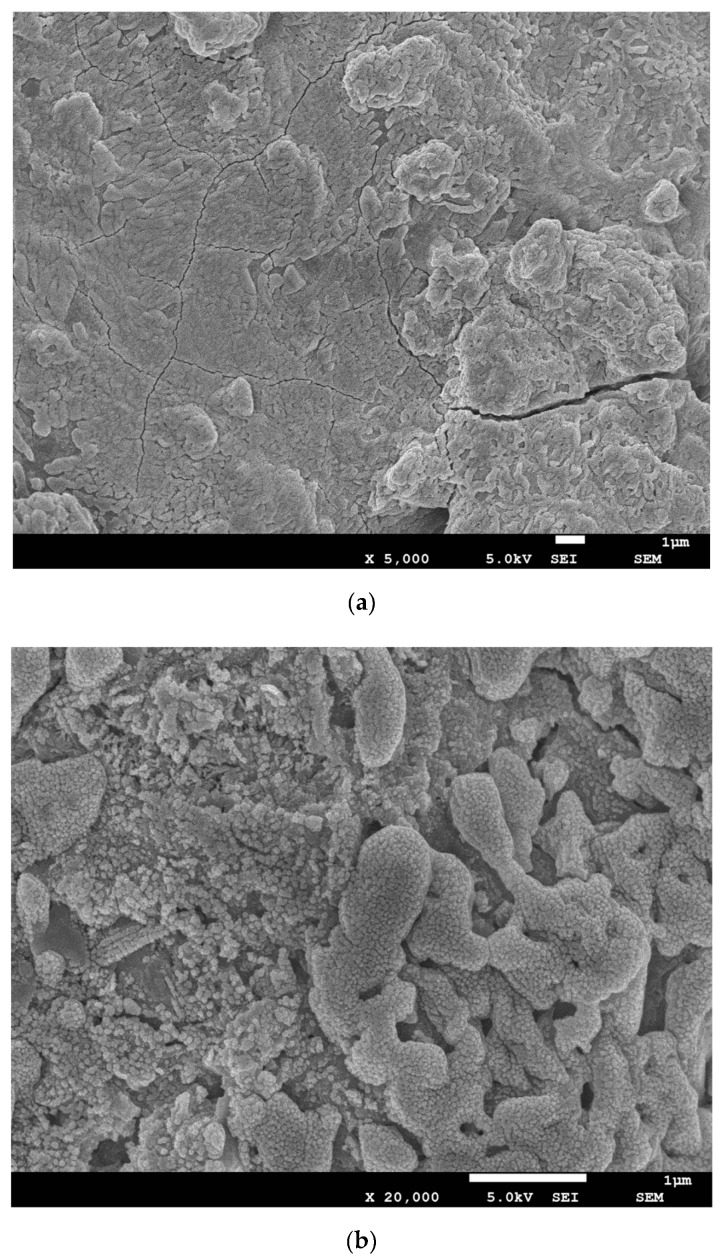
The microstructure of AAS after 28 days of curing. (**a**,**b**) AAS1, and (**c**,**d**) AAS2.

**Figure 7 materials-12-02089-f007:**
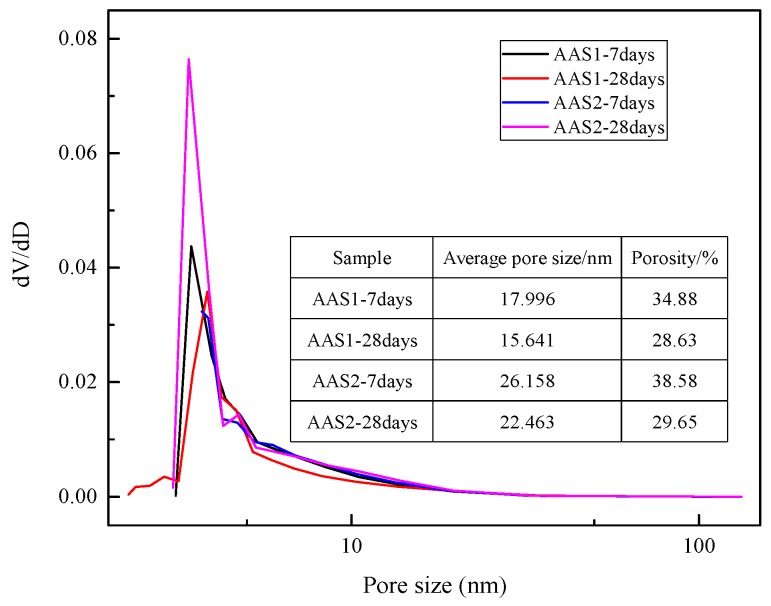
Pore size distribution of AAS.

**Figure 8 materials-12-02089-f008:**
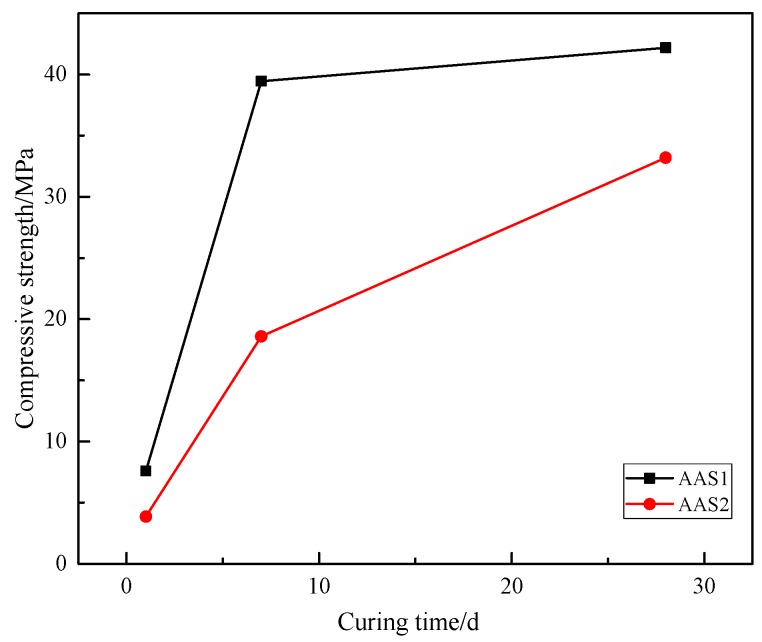
Compressive strength of alkali-activated slags.

**Table 1 materials-12-02089-t001:** Chemical composition of blast furnace slags (BFS) (wt.%).

Components	BFS1	BFS2
SiO_2_	32.53	31.12
Al_2_O_3_	16.12	9.74
CaO	38.01	11.02
Fe_2_O_3_	1.54	34.94
MgO	8.71	1.48
Na_2_O	1.33	1.32
K_2_O	1.25	1.83
MnO	1.81	1.97
SO_3_	1.25	1.28
Basicity index [(CaO + MgO)/(SiO_2_ + Al_2_O_3_)]	0.96	0.31
Hydration modulus [(CaO + MgO + Al_2_O_3_)/SiO_2_]	1.93	0.71

**Table 2 materials-12-02089-t002:** Characteristic particle diameters and BET surface areas of BFS.

BFS	D_10_ (μm)	D_50_ (μm)	D_90_ (μm)	BET Surface Areas (m^2^/kg)
BFS1	2.6	17.9	60.2	905.3
BFS2	2.8	18.9	57.9	873.4

**Table 3 materials-12-02089-t003:** Elemental analysis of AAS.

Sample	Concentration							Molar Ratios		
	Na	Mg	Al	Si	Fe	O	Ca	Ca/Si	Na/Al	Si/Al
AAS1	12.65	2.43	7.40	11.25	5.07	43.10	17.06	1.06	2.01	1.47
AAS2	10.85	—	3.63	10.01	28.67	38.10	6.76	0.47	3.51	2.66

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
