# Peer review of "Microstructural and Mechanical Properties of Alkali Activated Materials from Two Types of Blast Furnace Slags"

_materials, 2019, doi:10.3390/ma12132089_

Round 1
Reviewer 1 Report
This is a thorough report on alkali activation of two types of blastfurnace slag. The scientific experiments carried out are detailed enough and the article is presented in a logical way, although there is some restructuring needed. The level of English is poor and considerable editing is required, but the literature studied is adequate and the topic is presented well. My main concern is twofold:
- A lot of research is available on alkali activation of blastfurnace slag. Although the analytical tests are adequate, the novelty of the research needs to be pointed out.
- From the two BFS tested, the results vary greatly, mostly due to different iron oxide content. The effect of the iron oxide content on alkali activation needs to be discussed and elaborated more. The intrinsic properties of the difference in performance of the two slags need to be investigated further.
Author Response
Response to Reviewer 1 Comments
Point 1: A lot of research is available on alkali activation of blast furnace slag. Although the analytical tests are adequate, the novelty of the research needs to be pointed out.
Fe containing precursor materials and the role of Fe during the alkali activation are also currently receiving attention. J.S.J.van Deventer et al. deduced that reactive Fe from fly ash could reprecipitation much faster than Si and Al during geopolymerization according to the report by Valérie Daux et al. R.C. Kaze et al. studied the potential by using iron-rich laterite with kaolinite as precursor materials to synthesize inorganic polymers through alkaline activation. They found that the dissolved Fe3+ participated in reaction of geopolymerization and then inserted into the network matrix of aluminosilicate hydrate. Yong Hu et al. reported the geopolymer material synthesized from alkali-thermal pretreated Fe-rich Bayer red mud (RM) and fly ash and investigated the role of Fe species in geopolymerization by Mössbauer spectra. They found that the coordinated Fe3+ replaced Al3+ in the aluminosilicate structure of geopolymer. Rodrigue Cyriaque Kaze et al. developed building materials by applying the technology of geopolymerization using naturally abundant iron-rich laterite as raw material and rice husk ash. The enhanced mechanical properties of laterites based geopolymer composites resulting from the formation of Fe-silicates made them as promising candidates for applications in construction.
However, there are limited researches about the BFS containing high level of Fe and the distinct characteristics after alkali activation. The paper presents the preparation of AAS using two slags (one of them contains high level of Fe) with various properties. The performances are systematically compared and evaluated in terms of hydration kinetics, phase changes, FTIR analysis, thermal and mechanical properties, as well as microstructure.
Point 2: From the two BFS tested, the results vary greatly, mostly due to different iron oxide content. The effect of the iron oxide content on alkali activation needs to be discussed and elaborated more. The intrinsic properties of the difference in performance of the two slags need to be investigated further.
Please see the content in Section 4 Discussions.
The process of alkali activation process starts from the BFS dissolves to release Al3+, Si4+ and Ca2+ after contact with alkali activator solutions, indicated by the exothermic peak in Figure 2. The rate of the formation of hydration products impossibly independent on the leaching speed. As shown in Figure, it takes less time for AAS1 to reach the peak value, on the other hand, AAS1 enjoys a higher exothermic peak compared to AAS2, indicating that the hydration speed of AAS1 is faster. On one hand, BFS1 contains higher amount of basic oxides, such as CaO and MgO, which could react with water and generate Ca (OH)2 and Mg (OH)2, enhancing the amount of OH-. In the process of alkali activation process, OH- could accelerate the leaching speed of Al3+, Si4+ and Ca2+, which are block components of hydration products (i.e. C-(A)-S-H, N-A-S-H). On the other hand, as reported by Van Deventer et al., the active Fe from the BFS2 could reprecipitate as hydroxide or oxyhydroxide phases more rapidly than Al, Si or Ca ions, which will consume the OH-ions from the solution phase and therefore slowing down the dissolution of the remaining BFS particles.
The compressive strength of AAS highly depends on the amount of hydration product after the BFS being activated by alkali solution. As described above, BFS1 contains higher content of basic oxides, which can accelerate its dissolution and generate more hydration product, mainly as C-(A)-S-H, in the early age. This could lead to more heterogeneous matrix in the microstructure (see Figure 7 in section 3.5). However, the Fe content in BFS is higher, hindering the leaching of Al3+, Si4+ and Ca2+ ions, which are the main constituent part of the hydration product. Therefore, the hydration of BFS2 lasts for a long time, even after 27 days. This is why the growth rate of compressive strength for AAS2 is larger than that of AAS1 in the later age (see Figure 8 in section 3.6). Thermogravimetric analysis has been used to signify the amount hydration production in many studies. From Figure 5, the mass loss for AAS1 is about 5.4% before 200℃, mainly resulting from the loss of the adsorbed or interlayer water of C-(A)-S-H and the decomposition of ettringite. This number is larger than AAS2, whose mass loss is 1.9 approximately. The larger weight loss during this temperature period agrees well with the compressive strength testing result (see Figure 8) and microstructure analysis (see Figure 7). The larger formation of C-(A)-S-H and ettringite should be related to higher strengths and more compact matrix.
Although the compressive strength of AAS2 is lower than AAS1 in any curing age, AAS2 enjoys better thermal stability. The weight loss percentages of AAS1 and AAS2 are 14.9% and 3.6% after heating to 1000℃. As has been reported in the XRD analysis (see Figure 3 in section 3.2), AAS1 contains more C-(A)-S-H, hydrotalcite and calcium carbonate, which are not thermal stably in higher temperature. Dehydration, dehydroxylation and decomposition will take place when heated, leading to lager mass loss. While, by contrast, alkali activating BFS2 produces less such hydration products. On the other hand, BFS2 contains more Fe2O3 and Fe will inset into the hydration network ([-Fe-O-Si-O-Al-O-]) after being alkali activated. The EDX analysis (see Table 2 in section 3.5) also confirms this idea, which shows Fe content in the hydration product of AAS2 is 28.67%. Therefore, the results in this study reveal the prospect of BFS contains higher content of Fe to make alkali activated materials to heat resistance.
In general, although the BFS2 shows slow hydration rate being activated by alkali solution, it is possible to add some accelerator or preparation conditions to enhance the hydration rate. Similar studies are in progress in our research team.
Reviewer 2 Report
The paper deals with an examination and characterisation of alkali activated slags. The manuscript is written well, the used analytical methods are appropriate. The authors present a lot original experimental results and the conclusions are supported by the experimental observations. I have several comments to improve the manuscript as listed below:
1. Modify, please the Abstract, omit the notification of the samples S1, S2 etc. Change “laser particle analyser” to “laser granulometry” since you are reporting the methods used.
2. Line 36 – do you mean CO2?
3. Lines 55-58: I suggest using past tense. Modify the first sentence of the last paragraph in the Introduction: The following discussion will be focused … to The paper presents … The objective of the work is…
4. There is a need to exactly define the notification of the samples AAS1, BFS, S, etc.
5. Fig. 3 – XRD – what is the peak on AAS1, 7 days - at 21° of 2theta related with? Is that portlandite? Please specify the peak.
6. You reported very different mass losses of 14.9% and 3.6%, respectively, for the samples. How could be the higher stability of AAS2 explained?
Author Response
Response to Reviewer 2 Comments
Point 1: Modify, please the Abstract, omit the notification of the samples S1, S2 etc. Change “laser particle analyser” to “laser granulometry” since you are reporting the methods used.
We have modified it.
Point 2: Line 36 – do you mean CO2?
Yes, we mean CO2, not CO.
Point 3: Lines 55-58: I suggest using past tense. Modify the first sentence of the last paragraph in the Introduction: The following discussion will be focused … to The paper presents … The objective of the work is…
We have modified it.
Point 4: There is a need to exactly define the notification of the samples AAS1, BFS, S, etc.
We have notified in the place where they are mentioned in the first time.
Point 5: Fig. 3 – XRD – what is the peak on AAS1, 7 days - at 21° of 2theta related with? Is that portlandite? Please specify the peak.
It is not portlandite, portlandite does not have a peak in 21° according to the PDF 98-020-222. According to the peak place, it seems like a quartz peak. However, this peak does not be found in the raw material (BFS1). So, we contribute this peak to the impurities during the sample preparation.
Point 6: You reported very different mass losses of 14.9% and 3.6%, respectively, for the samples. How could be the higher stability of AAS2 explained?
The weight loss percentages of AAS1 and AAS2 are 14.9% and 3.6% after heating to 1000℃. As has been reported in the XRD analysis (see Figure 3 in section 3.2), AAS1 contains more C-(A)-S-H, hydrotalcite and calcium carbonate, which are not thermal stably in higher temperature. Dehydration, dehydroxylation and decomposition will take place when heated, leading to lager mass loss. While, by contrast, alkali activating BFS2 produces less such hydration products. On the other hand, BFS2 contains more Fe2O3 and Fe will inset into the hydration network ([-Fe-O-Si-O-Al-O-]) after being alkali activated. The EDX analysis (see Table 2 in section 3.5) also confirms this idea, which shows Fe content in the hydration product of AAS2 is 28.67%.
Reviewer 3 Report
This paper dealing with BFS composition effects on alkali-activated materials performances is interesting and well written. However, I have some minor comments that should be addressed:
Line 8: after alkali-activation or after being alkali-activated
Line 45: may vary instead od may various
Line 95: What are the BM mortars. Mix design should be clearly provided
Line 134: Starts instead of start
Line 195: I do not see compressive strength in figure 3
Author Response
Response to Reviewer 3 Comments
Point 1: Line 8: after alkali-activation or after being alkali-activated
We have modified it
Point 2: Line 45: may vary instead od may various
We have modified it
Point 3: Line 95: What are the BM mortars. Mix design should be clearly provided
We mean AAS samples, the BFS after being alkali activated. We have modified it in the text.
Point 4: Line 134: Starts instead of start
We have modified it
Point 5: Line 195: I do not see compressive strength in figure 3
This is a minor mistake, please see compressive strength in figure 8
Round 2
Reviewer 1 Report
The authors have pointed out the novelty of the paper and have also enhanced the discussion section
Author Response
Thanks for your help